# Efficient Cell Segmentation from Electroluminescent Images of Single-Crystalline Silicon Photovoltaic Modules and Cell-Based Defect Identification Using Deep Learning with Pseudo-Colorization

**DOI:** 10.3390/s21134292

**Published:** 2021-06-23

**Authors:** Horng-Horng Lin, Harshad Kumar Dandage, Keh-Moh Lin, You-Teh Lin, Yeou-Jiunn Chen

**Affiliations:** 1Department of Computer Science and Information Engineering, Southern Taiwan University of Science and Technology, Tainan 710301, Taiwan; hhlin@stust.edu.tw; 2Department of Electrical Engineering, Southern Taiwan University of Science and Technology, Tainan 710301, Taiwan; harshadkdandage@gmail.com (H.K.D.); chenyj@stust.edu.tw (Y.-J.C.); 3Department of Mechanical Engineering, Southern Taiwan University of Science and Technology, Tainan 710301, Taiwan; 4a414041@stust.edu.tw

**Keywords:** electroluminescence image, single-crystalline silicon photovoltaic module, cell segmentation, defect detection, pseudo-colorization

## Abstract

Solar cells may possess defects during the manufacturing process in photovoltaic (PV) industries. To precisely evaluate the effectiveness of solar PV modules, manufacturing defects are required to be identified. Conventional defect inspection in industries mainly depends on manual defect inspection by highly skilled inspectors, which may still give inconsistent, subjective identification results. In order to automatize the visual defect inspection process, an automatic cell segmentation technique and a convolutional neural network (CNN)-based defect detection system with pseudo-colorization of defects is designed in this paper. High-resolution Electroluminescence (EL) images of single-crystalline silicon (sc-Si) solar PV modules are used in our study for the detection of defects and their quality inspection. Firstly, an automatic cell segmentation methodology is developed to extract cells from an EL image. Secondly, defect detection can be actualized by CNN-based defect detector and can be visualized with pseudo-colors. We used contour tracing to accurately localize the panel region and a probabilistic Hough transform to identify gridlines and busbars on the extracted panel region for cell segmentation. A cell-based defect identification system was developed using state-of-the-art deep learning in CNNs. The detected defects are imposed with pseudo-colors for enhancing defect visualization using K-means clustering. Our automatic cell segmentation methodology can segment cells from an EL image in about 2.71 s. The average segmentation errors along the x-direction and y-direction are only 1.6 pixels and 1.4 pixels, respectively. The defect detection approach on segmented cells achieves 99.8% accuracy. Along with defect detection, the defect regions on a cell are furnished with pseudo-colors to enhance the visualization.

## 1. Introduction

With the advancements in renewable energy technologies, solar energy has been attracting more and more public attention in recent years. With the increasing demand for high-quality and efficient solar panels, the photovoltaic (PV) industries are facing challenges in the quality inspection of solar panels produced. The electro-luminescence imaging is a well-established technique in the PV industry to evaluate the quality and to identify damages to photovoltaic solar panel modules. A PV module is an assembly of photovoltaic cells, known as solar cells, arranged on a single frame for large-scale applications. PV modules consist of multiple electrically connected solar cells sealed in an environmentally protective laminate, and cells are the fundamental building blocks of PV solar panel systems. The EL technique provides images with very high-resolution details assisting the defect detection of fine-level microscopic flaws such as a crack and broken finger for automatic optical inspection using computers. An EL image is used to enhance the detection of defects with size, position, and orientation.

During PV module production, an EL imaging device can be used to record solar panel images at very high resolution, in which possible manufacturing defects in various sizes, positions, and orientations can be captured. Traditionally, the visual inspection of PV modules is carried out manually by trained experts/inspectors to identify defects and to rank quality levels. However, the microscopic flaws, such as cracks and/or broken fingers, cannot be easily and fully inspected by a human in a short time period in a real-time production line. Due to the lack of efficient and automatic defect analysis of PV panels, the time-consuming, manual inspection of selected EL images is still reluctantly conducted by field operators and is prone to unstable defect examination results by inexperienced inspectors.

Considering the requirement of PV industries for automatic quality inspection of PV modules using EL images, we propose a novel *"SCDD"*, automatic *Segmentation of Cells and Defect Detection* approach. SCDD is a method to extract cells from an EL image of single-crystalline silicon (sc-Si) PV module, detect defects on the segmented cells using deep learning and enrich defect regions with a pseudo-colorization method. An automatic cell segmentation method is based on the structural joint analysis of Hough lines. A defect inspection approach for cell images based on deep learning for practical applications is developed. Our experimental results show that the segmentation of individual cells is important in automatic defect identification for quality inspection of a PV module.

The results of our automatic and efficient cell segmentation approach are shown in Figure 1. A defected cell may contain abnormal regions, such as cracks (Figure 1a), and contamination defects (Figure 1b). Cracks on a PV module are caused by mishandling of a PV module, and contamination defects are caused by contamination of impurities during the manufacturing process. These defective cell images are manually labeled for training the classifier and detector.

We formulate our algorithms for automatic cell segmentation from an EL image of a PV module and defect detection on the segmented cells. The flowchart in Figure 2 exhibits the overall working pipeline of our proposed system. The workflow of the SCDD method comprises of following six steps.


Step 1: Image pre-processing to remove undesired noises from the original EL image by using Gaussian filtering.Step 2: Applying the contour tracing algorithm to identify contours and extract the required panel region.Step 3: Using probabilistic Hough transform to identify gridlines and busbars.Step 4: Segmentation of individual cells with the help of identified gridlines.Step 5: Defect detection on cell images by state-of-the-art deep convolutional neural networks.Step 6: The detected defects are enriched with pseudo-colors for enhanced visualization of defects.


The ultimate results of our proposed approach of cell segmentation and defect detection within bounding boxes including enhanced visualization of the defects by pseudo-colors are shown in Figure 3.

The features of the proposed SCDD approach include:The cells in an EL image of a PV module are segmented automatically for integrating CNNs with transfer learning [1] to detect defects on solar cells. The proposed cell-based defect detection module using YOLOv4 [2] obtains 99.8% accuracy and outperforms both the cell-based defect classification with ResNet50 [3] and the panel-based defect detection with YOLOv4 in the experiments.The proposed cell segmentation approach works accurately to localize the panel region from an EL image and to segment cells from the localized panel image. The segmentation method is simple and efficient as compared to the other cell segmentation techniques [4,5].We use a dataset consisting of 7140 solar cell images to perform an extensive evaluation of the proposed cell segmentation method. The proposed cell segmentation technique works efficiently with an average segmentation error of only 1.5 pixels.The detected defects are visualized with pseudo-colors to highlight the defect textures for better inspection. The pseudo-colorization uses K-means clustering on detected bounding boxes of defects. The defect localization with proposed pseudo-colorization on defects performs efficiently compared to the conventional digital image processing-based defect detection such as Gauss filtering [6] and $L_{o}$, gradient minimization with Fourier image reconstruction [7].

The remainder of the research article is organized as follows. A literature survey is mentioned in Section 2. The methodology of our proposed SCDD approach is discussed in Section 3. In Section 4, we present a detailed procedure of cell extraction from EL images, which is the core part of SCDD. Experimental results, including error analysis of cell extraction module and defect detection with pseudo-colorization, are discussed in Section 5. Finally, we conclude our research work in Section 6.

## 2. Related Work

The PV modules may contain defects that cause adverse degradation of performance. Kontges and their team [8] surveyed the various types of defects on PV modules depending on various environmental conditions leading to failure. Many PV modules are returned back to the manufacturers due to the failure of PV modules after they are deployed. Existing inspection of solar module performance often relies on the measurements of the physical properties of solar modules. For example, the evaluation of outdoor c-Si modules performance in extreme environmental conditions is proposed by Kahoul et al. [9] based on the I-V curve field measurements and visual inspections. Unlike [9], our defect detection approach is designed for inspecting the manufacturing quality of solar panels in a factory production line. As a stand-alone image-based inspection system, the measurements of I-V curves are not included in the proposed SCDD approach, owing to the fact that it may be generally difficult to quantitatively address the influence of small subtle defects on the I-V curves of a solar panel.

Automatic cell segmentation is an essential procedure for automating the visual inspection of EL images. Deitsch et al. [5] propose a robust automated segmentation method to rectify EL images by extracting the rigid edges. Individual solar cells were extracted from EL images of mono-crystalline and poly-crystalline PV modules by locating the gridlines and busbars. In the solar cell extraction procedure, factors affecting the extraction of boundaries are background noises and irregular positioning of the gridlines on the EL module images. Sovetkin and Steland [4] proposed an automatic cell segmentation approach to extract the cell area of single-cell mini-module. In their approach, cells can be extracted even in a noisy background, and the correction of perspective distortion is implemented by detecting the changes of cumulated sums (CUSUM).

Unlike [4], our cell segmentation is implemented in a structural hierarchy, where the coarse-level panel region is extracted from an EL image by contour tracing and the fine-level cell structures, such as gridlines and busbars, are identified by probabilistic Hough transform for further cell segmentation.

After the cells are segmented from EL images of PV modules, the very next important step is to find defects on cells for efficiency evaluation. Deitsch et al. [10] propose an automatic supervised classification of defective solar cells using a support vector machine (SVM). Using images of segmented cells, extraction of a set of descriptors is implemented and these descriptors are used to train a classifier to classify cells into defective or functional cells. Akram et al. [11] proposed an automatic detection of PV module defects in infrared images using deep learning techniques. Infrared (IR) imaging is performed on normal operating and defective PV modules to obtain a dataset of images. The authors developed a CNN architecture from scratch to classify IR images of PV modules. Tang et al. [12] proposed an efficient joint data augmentation approach for combining the image alternation and developed a GAN model for EL images to improve the performance of the machine learning models. A CNN-based model was developed for PV module defect detection and classification. Bartler et al. [13] proposed an automated classification of defected solar cell images with adapted VGG16 architecture by reducing the number of filters and the size of the fully connected layers to reduce the total number of parameters due to a smaller number of labeled training samples. Chen et al. [14] designed an optimized multi-spectral convolutional neural network classifier by adjusting the depth of a model and evaluating variation influenced by convolutional kernel size. A multi-spectral solar cell CNN network model is proposed to extract the multi-spectrum features of the solar cell surface. Parikh et al. [15] proposed a machine learning-based defect identification method in which classifiers such as, SVM, K-NN, and Random Forest are used. They present a method to extract statistical parameters from the histogram of cell images into feature vectors and utilizes them as a feature descriptor. These descriptors are used to train a classifier to classify the defect cells. Evaluation of PV modules using a CNN can also be implemented with the help of thermography images of PV modules. Haque et al. [16] proposed a fault diagnosis system by training the multi-layer perceptron on a dataset of thermal images of PV module containing defects and VI curves.

Many alternate methods can be used to train a neural network classifier such as image feature vectors extracted by wavelet coefficients using similarity distance algorithm proposed by Vetova et al. [17] and feature extractor as kernel extreme learning machine proposed by Sawssen et al. [18]. Our proposed approach utilizes a dataset of cell images, manually annotated to train a detector, and defect detection is implemented by generating feature maps in a hierarchical structure.

In our previous work of pseudo-colorization [19], we propose a unsupervised method of k-means clustering for marking potential defect pixels in pseudo-colors in an entire EL image for human visual inspection. Specifically, given a template of an EL image with pseudo-color labels on its defect regions, we impose the pseudo-colors to other greyscale EL images for highlighting defect regions. With respect to different defect types and image structures, the template feature clustering and pseudo-colors are transferred to a defected EL image, whereas, in the current research work, the pseudo-colorization scheme is updated to create two feature groups for foreground and background by k-means clustering based on their pixel intensity distributions. The cluster of pixels belonging to the foreground is imposed with pseudo-colors for better defect inspection.

Wang et al. [20] proposed an improved k-means clustering algorithm in which a weight calculation method for abnormal behavior was used to minimize the degree of abnormal behavior risk by extracting eigenvalues from a set of abnormal behavior. With the proposed pseudo-colorization method in the SCDD approach, we implement simple k-means to cluster foreground and background pixels of the defect area within the bounding boxes detected by a detector and providing pseudo-colors to the foreground pixels.

## 3. Methodology

In this section, we describe a novel methodology to extract individual cells from an original EL image. The segmented cells are fed into a CNN for defect detection. These defect-detected cells are further propagated to the pseudo-colorization model to apply pseudo-colors to defect textures for enhancing visual inspection of defects.

### 3.1. Cell Segmentation

In the proposed SCDD method, the cell segmentation procedure includes 5 steps for extracting cells from an EL image. Based on the main idea of finding the gridlines to segment all individual cells, first, from an original EL image (Figure 4a) of a PV module, the panel region (Figure 4b) is localized and extracted using contour tracing. Identification of the gridlines and busbars is implemented by plotting horizontal and vertical lines on the extracted panel region by using the Probabilistic Hough Line transform.

The Figure 4c shows gridlines (–o–) and busbars (–o–) plotted in red and blue colors, respectively, on the panel image. The rectangular grids generated from the red gridlines represent cell boundaries. Using the identified grids with red rectangular boundaries, horizontal strips are extracted (Figure 4d). Dividing each horizontal strip into cell images (Figure 4e) is an efficient treatment for accurate defect detection. The details of cell segmentation will be presented in Section 4.

### 3.2. Dataset Generation

After the segmentation procedure, all extracted cells have a unified resoluttion of 346×346 and are further enlarged to 416×416 in dataset generation for both defect classifier and detector learning. A dataset of cell images is generated to train deep learning models by manually labeling the segmented images into the Defect and NonDefect classes. For panel-based defect detection, we have prepared a dataset of 96 panel images for training and 23 images for testing. Since each panel image contains 60 sc-Si cells, the numbers of cell images are 5760 and 1380 for training and testing, respectively, as mentioned in Table 1. All the cell images in the dataset are manually labeled as *Defect* and *NonDefect* classes.

Due to the limited number of EL panel images given by the manufacturer, we artificially augment the dataset by horizontal and vertical flips to generate more data samples and to increase the data variability for training deep neural networks. The augmented cell image dataset used for the cell-based defect classification is shown in Table 2.

In contrast to the dataset preparation for cell defect *classification*, the data preparation for panel-based and cell-based defect *detection* is slightly different.

As not all cell images in Table 1 contain defects, the selection of cell image samples that have various defects from the dataset of Table 1 is required. As listed in Table 3, 1257 and 1380 representative cell samples containing defects are chosen for training and testing of cell-based defect detector, respectively. The defects on the chosen cell images are manually annotated with bounding boxes to train and test the detector.

### 3.3. Defect Detection

Segmented cell images (Figure 5a) are sent to the defect detection system. If any defect is detected on the cells (Figure 5b), it is further propagated to the pseudo-colorization system. Supervised deep learning with a deep convolutional neural network is widely adopted in defect detection applications. This research work involves a comparative study between a *classifier* and a *detector* to identify defects on cells.

In the two-class cell classification, the economic and yet powerful ResNet50 [3] is adopted, whose computational structure in the implementation is shown in Table 4. The five main CNN layers from 173×173 to 6×6 are used to generate feature maps in a hierarchical structure for defect identification.

Our proposed SCDD approach uses YOLO to detect defects on the cell images. YOLO is a powerful single convolutional network that simultaneously predicts multiple bounding boxes and class probabilities for those boxes. For the performance comparison of different YOLO versions, the defect detection is implemented with YOLOv3 [21] and YOLOv4 [2].

Due to the better performance of YOLOv4 in later experiments, the computational architecture of YOLOv4, as a successor of YOLOv3, is in particular introduced in Figure 6. The YOLOv4 is made of a backbone, a neck, and a head module. The backbone module of YOLOv4 is composed of CSPDarkNet53 [22].

The neck of the detector is a combination of a spatial pyramid pooling (SPP) layer and path aggregation network (PAN). The SPP [23] allows the generation of fixed-sized features irrespective of the size of feature maps through the pooling calculation. The PAN [24] allows better propagation of layer information from bottom to top and top to bottom. The YOLO head performs dense prediction, which comprises a vector containing the coordinates of the predicted bounding box, the confidence score of the prediction, and the label.

The images in our train and test dataset are labeled manually with bounding boxes using a labeling tool to generate XML files of annotations. Later on, these XML files are converted to YOLO format annotation files. Image annotation has a critical role in computer vision. The goal of image annotation is to assign relevant, task-specific labels to images. Data annotation requires a high level of domain knowledge, and we labeled defects on solar cells with the profound observation of defect regions of cracks and contamination defects as shown in Figure 1.

### 3.4. Pseudo-Colorization

The defects within detected bounding boxes are further marked with pseudo-colors for better visual inspection. The procedure of furnishing defect regions or textures with colors is regarded as pseudo-colorization (Figure 5c). Although pseudo-colorization has been explored in computer graphics applications, it is rarely applied to defect inspection of solar cells. In particular, we develop a low-complexity algorithm of defect colorization based on k-means clustering for distinguishing foreground (Defect) pixels from background (NonDefect) ones. The defects in detected bounding boxes are imposed with pseudo-colors for further human justification of bounding box detections and for easier observation of specific defect textures.

In Algorithm 1 of the proposed pseudo-colorization, we set k=2 for k-means clustering to extract foreground and background clusters from each bounding box of defect identified by a detector in a gray-scale cell image. The k-means clustering is embodied based on the pixel intensities in the detected bounding box. The pixels of the foreground cluster with a lower mean intensity value are imposed with pseudo-colors. Specifically, the pseudo-colorization on detected defects is enforced by (a) obtaining the dimensions of a bounding box drawn on a cell image (x_left and y_top coordinates and width and height) detected by YOLOv4 in the testing phase, and (b) cropping the area of cells within a bounding box as a region of interest (RoI) $R_{i}$ for further colorization with Algorithm 1. As a result, the pixels of foreground cluster of defect are colored red.
**Algorithm 1** Pseudo-colorization.1. I = $\left\{I_{1},I_{2}\right\}$ = k-means($R_{i}$, k = 2),              where I is a set of two intensity means $I_{1}$ and $I_{2}$, with $I_{1}<I_{2}$, and $I_{1}$ corresponds              to the foreground (defect) cluster.              $R_{i}$ is a bounding box of detected defect RoI in a cell image.2. Imposing a pseudo-color (red) to the pixels of the $I_{1}$ cluster in the RoI bounding box.

Besides the adopted k-means clustering, other clustering algorithms, such as mean-shift clustering [25], can also be used for pseudo-colorization. Due to the simplicity and the stable clustering output of k-means clustering, we choose it as the default implementation of pixel intensity clustering in Algorithm 1. Experimental comparisons of different colorization techniques of defect pixels will later be presented in the experiments.

## 4. Cell Segmentation

We develop five significant steps to extract cells from an EL image. Concerning the pipeline, in Figure 4, the cell segmentation steps propagate from extraction of panel region from an original EL image of PV module to gridlines and busbars identification, revealing borders of a cell, henceforth assisting in individual cells extraction. A detailed step-wise cell extraction methodology is discussed later in this section.

Step 1: Panel Region Extraction

The first and foremost step in solar cell segmentation is to extract the panel region from an EL image that represents an original solar panel. Here in this step, the appropriate panel region (Figure 7b) is extracted from an original EL image of a PV module (Figure 7a). The contour tracing algorithm is employed to extract a panel region, hence serving as an essential step for segmenting cells.

In Algorithm 2 of panel region extraction, an original EL image of resolution M×N pixels is converted to a greyscale image ($I_{b}$). The greyscale EL image is blurred with the Gaussian smoothing to reduce unwanted noises from an EL image.

The next step is to find contours for extracting the panel region. An adaptive threshold is adopted using the OTSU thresholding to discriminate panel region from the background in an EL image. Figure 8a shows the result of thresholding on an original EL image. Contour tracing is applied to find all contours on this thresholded image, and contours with the largest area are selected to extract the panel region. The contour area enclosed in a red box as shown in Figure 8b is the target area of the panel to be extracted. Figure 7b is the final extracted panel region.
**Algorithm 2** EL panel region extraction.1. Noise reduction using Gaussian blur, i.e.,            $I_{b}$ = GaussianBlur(*I*),            where, *I* is a greyscale EL image.2. OTSU thresholding, i.e.,            $I_{B}$ = OtsuThresholding($I_{b}$),            where, *B* is a binary image from the otsu threshold $T_{h}$.3. Contour tracing, i.e.,            C = FindContour($I_{B}$)            $C_{i}$ = { $C_{1}$, $C_{2}$, ....., $C_{n}$}            where, *C* is a set of *n* contours and                  $C_{i}$ is an ordered list of connected contours points.4. Panel region identification, i.e;            $i^{*}$ = MaxArea($C_{i}$) and            [*x, y, w, h*] = BoundingBox($C_{i^{*}}$)            where, *x* and *y* are top left x and y coordinates of bounding box, respectively.            *w* and *h* are width and height of bounding box, respectively.5. Panel region segmentation with the dimensions.

Step 2: Horizontal and Vertical Lines Plotting

To trace horizontal and vertical lines on the localized panel region, we apply the Canny edge [26] detection approach followed by the probabilistic Hough line technique [27], supported by an adaptive threshold value calculated with the overall intensity of an EL image.

In Algorithm 3, for horizontal and vertical lines plotting, we used the Canny edge detection to identify the edges of the cells on extracted panel region and probabilistic Hough transform to identify and plot lines on the extracted boundaries. We apply adaptive high and low thresholds based on the combination of Otsu thresholding and overall pixel intensity of the extracted solar panel region image. Otsu thresholding is applied to the panel image because it separates dark and light regions by iterating through all the possible threshold values and calculating a measure of spread for the pixel that falls in either the foreground or the background. To identify the actual edges on a panel image, we apply threshold values, high ($T_{h}$) and low ($T_{l}$). Any edges with an intensity gradient greater than $T_{h}$ are considered to be edges, and those below $T_{l}$ are discarded considering to be non-edges. To automatically select a $T_{h}$ threshold value, we apply a combination of Otsu thresholding on the panel image $I_{p}$ and the average intensity $\bar{I}$. As a result, the threshold values achieved have the ability to improve the accuracy of the edge detection with minimal noise. The Canny edge detection is used to locate edges of the panel image by employing these two adaptive thresholds. Figure 9a shows the result of Canny edge on the panel image required for Hough line estimation.
**Algorithm 3** Horizontal and vertical lines plotting.1. Threshold calculation, i.e.,            $T_{h}$ = OtsuThresh($I_{p}$) $-(\bar{I}/2)$            $T_{l}$ = (OtsuThresh$\left(I_{p}\right)$/2) $-(\bar{I}/2)$            where, $T_{h}$ and $T_{l}$ are the high and low thresholds, respectively            $I_{p}$ is extracted panel region,            $\bar{I}$ is Average pixel intensity of image $I_{p}$.2. Edge detection, i.e.,            $I_{E}$ = Canny($I_{p}$, $T_{h}$, $T_{l}$)3. Lines Estimation, i.e.,            Lines = HoughLines($I_{E}$, $T_{h}$)

Further, this image is used to estimate lines by using the probabilistic Hough line transform technique, and these lines are plotted on the solar panel region image. Figure 9b shows the horizontal and vertical lines plotted on the panel image.

Step 3: Plotting Edge Points on Panel Region

Subsequently, after obtaining the horizontal and vertical lines, all the edge points on the solar panel are marked. These edge points ultimately represent the boundaries of cells, thus assisting in segmenting an individual cell. To locate the edge points, the intersection points of all the horizontal and vertical lines are traced and plotted on the panel region image. Figure 10a shows the intersection points of the horizontal and vertical lines, and Figure 10b shows the location of all the edge points, representing the borders of each cell.

Step 4: Gridlines and Busbars Identification

Once all the edge points are identified, the next step is to find rectangular gridlines representing the actual boundaries of a cell. The gridlines are identified in two steps. First, all the edge points are identified and located along the four boundaries of an EL panel image. Second, the edge point is connected with lines along the same row and column. The first edge point $P_{1}$ is located, and further, with the help of the first edge point, all grid points located along all the boundaries are traced. Figure 11a shows the labeled boundary edge points raging from $P_{1}$ to $P_{32}$.

The edge points on the top boundary are extracted and denoted as a set of points $E_{T}$ that includes edge points $P_{1}$ to $P_{11}$. Similarly, edge points on the right, bottom and left boundaries are extracted and denoted as $E_{R}$ ($P_{11}$ to $P_{17}$), $E_{B}$ ($P_{17}$ to $P_{27}$), and $E_{L}$ ($P_{27}$ to $P_{1}$), respectively. The edge points circled in red in Figure 11a are the required edge grid points. Edge points on the top ($E_{T}$) and bottom ($E_{B}$) boundaries can be traced easily, whereas the left ($E_{L}$) and right ($E_{R}$) boundaries are challenging because $E_{L}$ and $E_{R}$ may contain intersection points of busbar lines. $E_{R}$ and $E_{L}$ edge points are traced with a minimum skip distance of β pixels between neighboring points.

To locate the gridlines, horizontal and vertical lines are drawn on the panel image by using a line drawing function to connect the extracted grid edge points alone to the respective row and column. In Figure 11b, the lines in red (–o–) are the gridlines representing the boundaries of individual cells and the lines in blue (–o–) represent the busbars. The numbers of horizontal gridlines, vertical gridlines, and busbars are 7, 11, and 24, respectively.

Step 5: Cell Segmentation

Segmentation of cells is performed in two steps. First, with the help of gridlines extracted in step 4, the panel image is segmented into 6 horizontal strips, consisting of 10 cells in each strip. The horizontal strips are the rectangular strips extracted from a solar panel image. As mentioned in Algorithm 4, **$T_{l}$**, **$T_{r}$**, **$B_{l}$**, and **$B_{r}$** are the four coordinates of a rectangular strip. Figure 12a shows the segmented horizontal strips. Second, to extract the cells from the horizontal segmented strip, the location of the next grid boundary coordinate is select as a threshold value of “α” pixels. Figure 12b shows the individual segmented cells. The number of cells segmented from an EL image is 60.
**Algorithm 4** Cell Segmentation.1. Horizontal strips extraction, i.e.,            $Strip_{s}$ = Segment(**$I_{p}$**, **$T_{l}$**, **$T_{r}$**, **$B_{l}$**, **$B_{r}$**)            where, *s* ranges from 1 to 6,                  **$T_{l}$** is Top-left edge, and **$T_{r}$** is Top-right edge,                  **$B_{l}$** is Bottom-left edge, and **$B_{r}$** is Bottom-right edge2. Cell segmentation, i.e.,            $Cells_{c}$ = Segment($Strip_{s}$, width, height)            where, *c* ranges from 1 to 10,                  width is the α number of pixels,                  height is the height of each *strip*.

The entire cell segmentation process takes 2.71∼2.81 s, depending upon the size of an input EL image. To train a defect detector, specific defect regions need to be further marked in bounding boxes in each cell image at its original size.

## 5. Experimental Results

The cell segmentation errors are evaluated to justify the effectiveness of our *SCDD* system in Section 5.1. Subsequently, the experiments of defect detection and visualization on cell images are presented in Section 5.2 and Section 5.3, respectively.

### 5.1. Segmentation Error Analysis

The errors of our cell segmentation results for segmenting 7140 cells along the x-direction and y-direction are calculated using:(1)$$Err_{x}=\frac{1}{N}[\sum_{i}^{n}|E_{x}\left|\right]\;and\;Err_{y}=\frac{1}{N}[\sum_{i}^{n}|E_{y}\left|\right]$$
where, *N* = Total number of cells, *n* = Number of cells in each EL image of module, *Ex* = Error along x-direction, *Ey* = Error along the y-direction.

The true cell boundary is labeled in green, and the extracted cell boundary is labeled in red, as shown in Figure 13. In our experiments of cell segmentation, distortion of 1 or 2 pixels are observed in some cells. The segmentation errors of our cell segmentation approach for the dataset of 7140 cells extracted from 96 EL images are about 1.6 pixels and 1.4 pixels in the x-direction and y-direction, respectively.

### 5.2. Defect Inspection

The performances of automatic defect inspection are evaluated at two levels: the panel-based defect detection and the cell-based defect identification. The panel-based defect detection is done with a detector, YOLOv4. The cell-based defect identification is accomplished with a classifier and two versions of the detector for performance comparisons. We used ResNet50 as a classifier and YOLOv3 [21] and YOLOv4 for performance comparison of the detectors. The organization of experiments shown in Figure 14 demonstrates the experiments conducted in our work. Both the panel-based and the cell-based defect detections are implemented and compared in terms of accuracy and efficiency to determine which processing unit, i.e., panel or cell, is more applicable to the task of quality inspection for the manufacturing industry.

Briefly, the results of experimental comparisons demonstrated in this section validate that the proposed cell-based defect detection approach is superior to the panel-based approach. In the comparisons of cell-based defect classification and detection, the derived cell defect detector from YOLOv4 gives the best performance.

#### 5.2.1. Cell-Based Experiments

In this section, we present the results of defect classification with ResNet50 and the results of defect detection with YOLOv3 and YOLOv4 on segmented cell images.

##### Cell-Based Defect Classification

The ResNet50 CNN model is trained with the dataset summarized in Table 2 to classify cell images into *Defect* and *NonDefect* classes. The specific structure of the ResNet50 implementation for this experiment is shown in Table 4. During the training process of the classifier [28], we freeze the weights of the first three layers of the original pre-trained ResNet50 network, and the remaining layers are trained. We added one more FC layer for converting 2D feature maps to the desired 1D output to assis with the need for two output nodes for our defect cell classification application to classify a cell into a defect or non-defect class. The Adam optimizer is used to fine-tune the pre-trained model. The batch size is set to 32, the initial learning rate is set to 1× 10${}^{-6}$, and the exponential decay of learning rate is set to 1× 10${}^{-2}$ after every 10 epochs, and the max number of epochs is 200.

As a result, the fine-tuned deep learning model can successfully identify cells with tiny defects that are difficult to be detected manually by human inspection. The confusion matrix of the resulting classification model is presented in Table 5. In experimental results, the precision and recall of the model are 99.57% and 99.15%, respectively, which proves the reliability of our deep learning classification model. Figure 15 shows some examples of cell image classification results.

##### Cell-Based Defect Detection

Although the results of the classifier are promising, field operators may not be able to know where the defect is located in each cell image and why the classifier determines the cell image as a defect. Operators need to pay extra attention to justify classification results. Thus, we also employ defect detection models to locate defects on cells. The defect detection is tested on EL panel-based images and on specifically segmented cell-based images. Table 3 shows the number of images used for training and testing for both the panel-based and the cell-based defect detection approaches.

The proposed cell-image-based defect detection is implemented with YOLO. For comparing the performances of two versions of the detector, defect detection is carried out with YOLOv3 and YOLOv4. The training parameters for YOLOv3 and YOLOv4 are kept similar, and we set an initial learning rate to 1× 10${}^{-3}$ with a decay rate of 5× 10${}^{-4}$ and a maximum number of iterations of 6000. We use an IoU threshold of 0.5 for evaluating and comparing the accuracy of the YOLO models.

In the testing phase, the mAPs for YOLOv3 and YOLOv4 are 71.6% and 98.5%, respectively. Adding up to the defect detection accuracy, the F1-score for YOLOv3 is 77.47 and that for YOLOv4 is 97.86. Defects detected on the cell images with YOLOv3 and YOLOv4 are shown in Figure 16. Although the defects in Figure 16a,b can be detected by YOLOv3, some bounding boxes of YOLOv3 do not completely cover the defect areas. On the other hand, as shown in Figure 16d,e, the corresponding detection results of YOLOv4 perfectly capture the defects. Moreover, the defects in Figure 16c are misdetected by YOLOv3, but are correctly detected by YOLOv4 in Figure 16f.

The dataset used to test the detector as mentioned in Table 3 contains 308 images. Among these 308 cell images in the test, 302 cells are identified as the ones containing defect bounding boxes by both the YOLOv3 and YOLOv4 detectors. Only six cells are not detected, as shown in Figure 17, probably owing to the tiny defect sizes. The experimental results of YOLOv3 and YOLOv4 differ in the confidence score of defect detection, as well as the precision of bounding box localization in the defect region. As shown in Figure 18, the defect detection of YOLOv4 (Figure 18b) is better than that of YOLOv3 (Figure 18a), where, in YOLOv4, the confidence scores of defected defects are higher and the resulting bounding boxes over the defects are more accurate.

#### 5.2.2. Panel-Based Experiment

For the comparison of panel-based and cell-based defect detection performances, a whole EL image is fed to a YOLOv4 detector in the panel-based approach in which all the defect regions of training images are manually labeled. The original resolution of panel images is 2450×1300. For YOLOv4, panel images are re-scaled to 1730×1048. The YOLOv4 model is trained with an initial learning rate of 1× 10${}^{-5}$ with a decay of 5× 10${}^{-4}$ and maximum steps of 6000. An intersection-over-union (IoU) threshold of 0.5 is adopted to measure the accuracy of the trained detector. In the testing phase, the mean average precision (mAP) is only 61.3%. The defect of the EL images can be still detected by the trained model, but the overall precision is 51% and the recall rate is also low at 69% only. Figure 19 shows the snapshots of defects detected on the EL images.

As shown in Table 6, we implement panel-based and cell-based defect classification and detection. The detection results of the panel-based approach are unsatisfactory due to the difficulty in recognizing the tiny defect regions on an entire panel image. The cell-based classification results are reliable, but defects within a cell cannot be clearly specified, thus making further manual inspection tough. In comparison with YOLOv3 and YOLOv4, the cell-based defect detection results of YOLOv4 are better than those of YOLOv3, because the hyper-parameters, such as anchor sizes, for training YOLOv4 can be optimally determined by auxiliary modules in the YOLOv4 framework.

To sum up, the proposed cell-based approach using YOLOv4 for defect detection (97.86%) outperforms the panel-based defect detection (58.65%) with YOLOv4, the cell-based defect classification (99.39%) with ResNet50, and the cell-based defect detection (77.47%) with YOLOv3.

Regarding the further comparison of selecting a panel or cell processing unit in defect detection, the cell-based defect detection gives a far better detection accuracy than the panel-based counterpart, as shown in Table 6, under acceptable takt time of cell defect detection for the industrial application of solar panel manufacturing. Precisely, the cell-based defect detection approach takes ∼3.12 s to inspect 60 cells in detail in a panel image, while the panel-based counterpart takes only ∼48 ms.

However, due to our preliminary, non-optimized implementation of cell segmentation using Python, in the proposed SCDD approach, the ∼2.22 s time consumption of cell segmentation dominates the whole ∼3.12 s takt time. The rest of cell defect detection for all the 60 cells takes only ∼900 ms (15 ms/each cell), which still meets the requirements for online application of the solar panel. The panel-based and cell-based approaches reflect the two design choices, emphasizing computational efficiency and accuracy, respectively. Further optimization of cell segmentation using C/C++ programming is doable for enhancing the processing speed and will be included in future work.

### 5.3. Pseudo-Colorization on Defect Region

Despite the promising performance of defect detection on cell images using YOLOv4, the detection results in bounding boxes are sometimes hard to verify by human inspection, because faint and small defects cannot be easily perceived by the naked eye. Further marking of defect pixels automatically shall be helpful for double-checking of defects and analysis by field inspectors to visually examine single-crystalline silicon PV modules. Therefore, we propose the pseudo-colorization of defects, as shown in Figure 20 and Figure 21, to enhance the visual observability of defects and validate the effectiveness of this post-processing step of fine-level defect identification by experiments.

A broad comparison of conventional image processing methods and two clustering techniques, i.e., k-means and mean-shift clustering, for pseudo-colorization of defect pixels in bounding boxes and cell images is conducted in the experiments. Note that both the k-means and the mean-shift clustering for pseudo-colorization are implemented by referencing the sample codes and the default parameter settings suggested in the documents of the SciKit-Learn library in Python. The similarities among the pixels in a bounding box are measured by the gray-scale intensities for two-group (foreground-and-background) clustering. The results of the pseudo-colorization comparisons are shown in Figure 20.

As shown in Figure 20b, the Otsu thresholding is applied to a whole cell to generate the global binarization result as a baseline. The results of global binarization contain additional textures of busbars and cannot locate defect regions properly. Distinguished from the global binarization, the coloring of local edge points on each defect bounding box is performed by Canny edge detection and is presented in Figure 20c. While some edge pixels on defect regions can be labeled, the defect region contours cannot be completely outlined yet. The defect colorization results derived by the Canny edge detection are apparently not clear enough for human visual inspection.

On the other hand, the pseudo-colorization by clustering techniques enhances the localization and the visualization of defects for manufacture operators. The defect coloring results by the mean-shift and the k-means clustering are depicted in Figure 20d,e, respectively. While both the clustering methods catch similar defect regions, the results of k-means clustering is slightly better in defect coloring accuracy and in visualization clarity. Moreover, the k-means clustering has fewer parameters and is more efficient in computation than the mean-shift clustering, which is advantageous to our application.

By displaying the defect bounding boxes and the defect region colorization side by side, as shown in Figure 21, field inspectors can easily identify defects, especially for dim and/or small ones. The time cost and human efforts on defect double-checking can be largely reduced. The manufacturing process of solar panels can thus be sped up by the automation of the SCDD approach.

## 6. Conclusions

Motivated by the requirement of automatic quality inspection of EL images of single-crystalline silicon solar panel images, we propose an SCDD approach to automatically segment cells, to detect the defects on segmented cells, and to apply pseudo-color to detected defects for better visualization. The proposed cell segmentation approach works accurately to extract the panel region from an EL image and to segment cells from the localized panel image. The proposed cell segmentation technique possesses small segmentation errors of only 1.6 pixels in the x-direction and 1.4 pixels in the y-direction. In the experimental comparisons, the proposed cell-based approach using YOLOv4 obtains 98.5% accuracy and outperforms both the cell-based defect classification with ResNet and the panel-based defect detection with YOLOv4.

Some assumptions and observations concerning the proposed SCDD approach are further discussed here. First, in solar cell segmentation, we assume that the size of solar cell is in a typical range, e.g., about α pixels in cell width, given a fixed camera capturing setting. Such typical parameters of cell size in width and height are applied to the selections of cell corner points in Algorithm 4. Furthermore, the panel structure consisting of gridlines and busbars is commonly seen and referenced in the cell segmentation. The cues of cell size and prominent edges on solar panels, originating from the inherent panel structure and the inspection camera settings, are reasonably incorporated in our algorithm design.

Second, the (mean) intensities of solar cells in a panel image are highly uneven, as shown in Figure 2. Therefore, the panel-based defect detection suffers from not only the problem of small defects relative to a large panel but also the challenge of varying intensity distribution among cells. Based on the relatively smooth intensity distribution in cell pixels, detecting defects in a cell is indeed more effective.

Third, the irregular textures of cracks and broken fingers are perceivably distinct from regular cell structures of busbars, which is a justifiable observation/assumption in defect identification. While the experimental deep CNNs of classifiers and detectors are all capable of capturing such discriminative defect features through data learning, they might fail to detect tiny defects around cell boundaries.

Fourth, the coloring of defect pixels in a solar cell is based on the presumption that defects are usually darker in EL imaging. Although intensity binarization can be typically used to filter out dark regions from a whiter background, the proposed k-means clustering still gives finer pixel coloring results than simple binarization without largely increasing the computational complexity.

The proposed cell segmentation technique possesses small segmentation errors of only 1.6 pixels and 1.4 pixels in the x- and y-directions, respectively. In the experimental comparisons of defect classification and detection, the proposed cell-based defect detection approach using YOLOv4 obtains 98.5% accuracy and outperforms both the cell-based defect classification with ResNet and the panel-based defect detection with YOLOv4. The experimental results validate that the proposed SCDD approach, including cell segmentation and cell-based defect detection with YOLOv4, is highly accurate and reliable for the automatic defect identification during PV module manufacturing. Furthermore, by incorporating the post-processing of defect colorization by the simple and efficient k-means clustering, the efficiency of defect visual inspection by operators in selective double-checks of defects can be largely enhanced.

The proposed SCDD approach has the advantages of (1) applying firstly the cutting-edge deep CNNs to the practical and important application of solar panel production, (2) integrating useful computational heuristics to the pre-processing of EL images for stable and precise cell segmentation, (3) verifying the importance and effectiveness of defect detection on cell units, (4) achieving highly accurate defect detection rates by deep learning from limited training samples of cell images, and (5) incorporating the pseudo-colorization of defects as an elegant post-processing step for better defect visualization beyond the conventional defect detection. Nevertheless, some deficiencies of the SCDD approach can still be observed and will be included in future work, including the code optimization for speeding up cell segmentation, further tests of defect detection on double- or quadruple-cell unit for balancing the defect detection accuracy and efficiency, and the improvement of tiny defect detection around cell boundaries in particular.

## Figures and Tables

**Figure 1 sensors-21-04292-f001:**
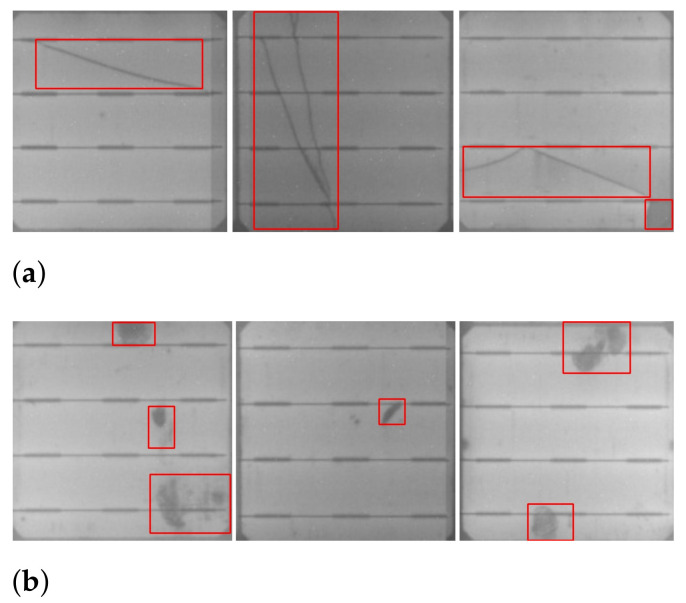
Samples of segmented solar cells containing defects: (**a**) cracks, (**b**) contamination defects.

**Figure 2 sensors-21-04292-f002:**
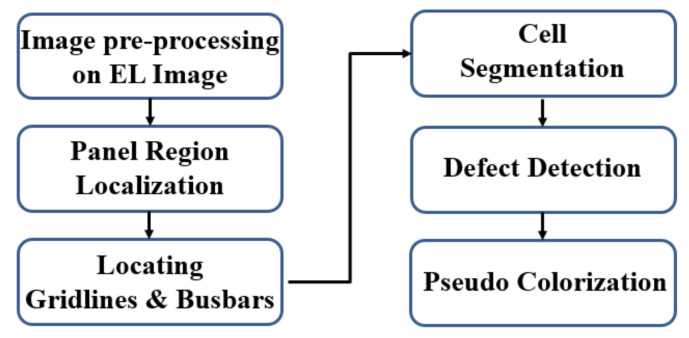
Flowchart of the SCDD method.

**Figure 3 sensors-21-04292-f003:**
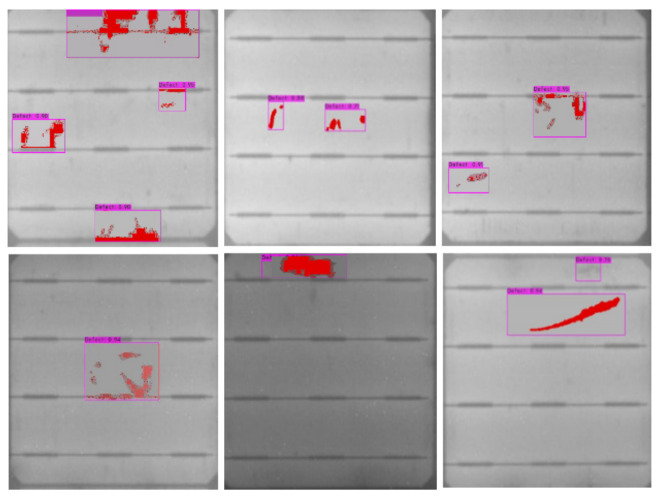
Results of the SCDD model.

**Figure 4 sensors-21-04292-f004:**
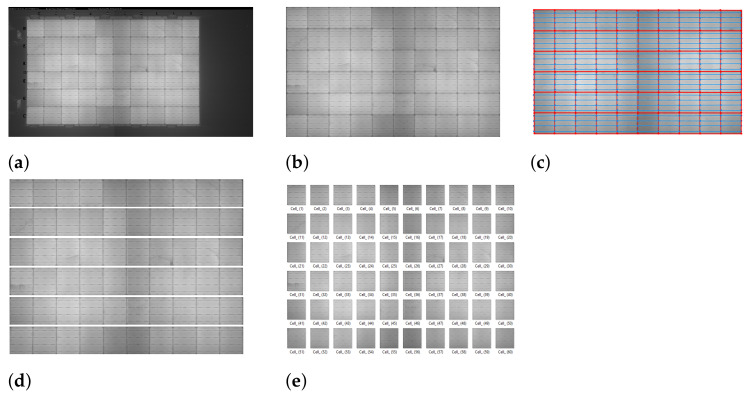
Pipeline for automatic cells segmentation in SCDD approach. (**a**) Original EL image of the PV module. (**b**) Panel region localization. (**c**) Gridlines and busbars identification. (**d**) Segmenting horizontal strip regions. (**e**) Segmenting individual cells.

**Figure 5 sensors-21-04292-f005:**
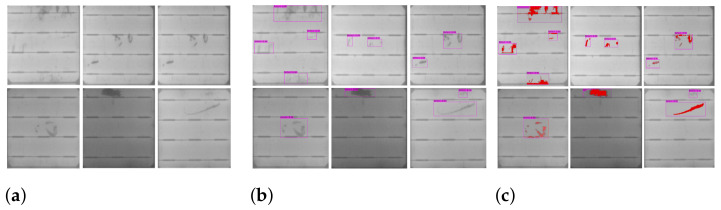
Procedure for defect detection and pseudo-colorization. (**a**) Selecting segmented cells for defect detection. (**b**) Defects detected by detector. (**c**) Pseudo-Colorization of defects.

**Figure 6 sensors-21-04292-f006:**
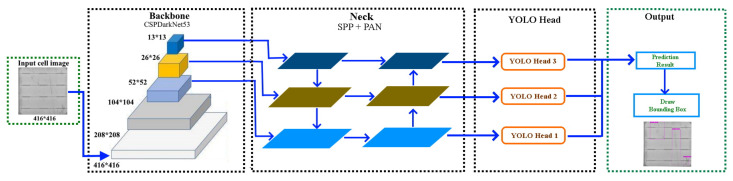
The architecture of YOLOv4.

**Figure 7 sensors-21-04292-f007:**
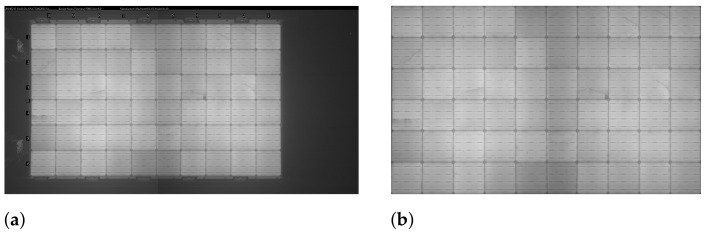
Panel region extraction, (**a**) original EL image, (**b**) extracted panel region.

**Figure 8 sensors-21-04292-f008:**
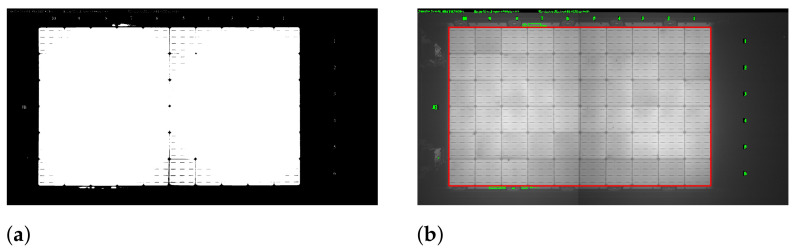
Contour tracing. (**a**) Thresholded EL image. (**b**) Target contour area for panel region extraction.

**Figure 9 sensors-21-04292-f009:**
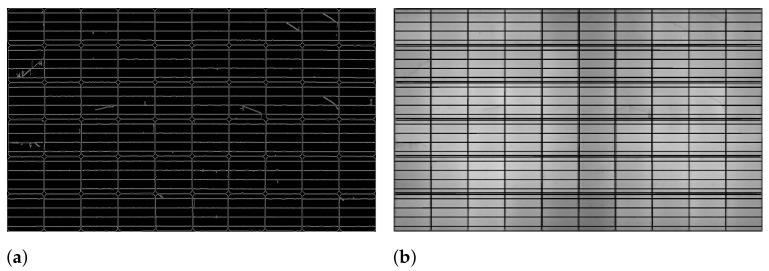
Horizontal and vertical lines estimation. (**a**) Edges of grids. (**b**) Horizontal and vertical lines on the panel image.

**Figure 10 sensors-21-04292-f010:**
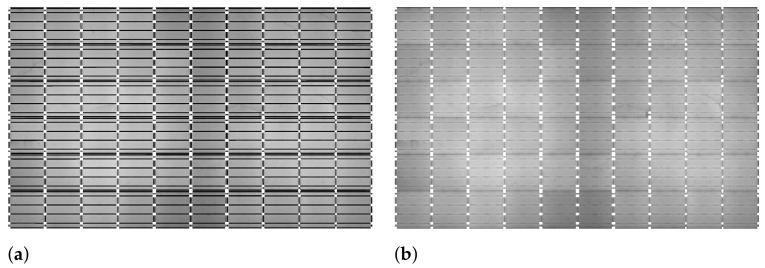
Plotting edge points on panel region. (**a**) Intersection points of horizontal and vertical lines. (**b**) Plotting intersection points on the panel region of EL image.

**Figure 11 sensors-21-04292-f011:**
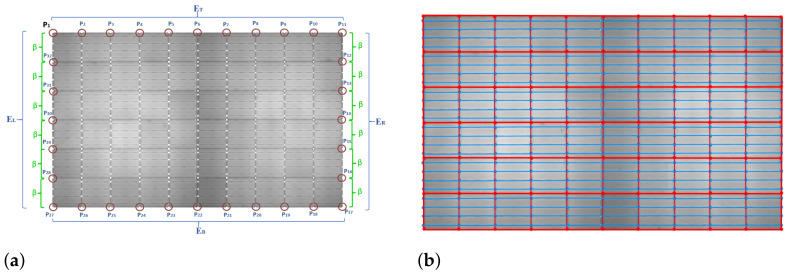
Gridlines and busbars identification. (**a**) Location of boundary edge points. (**b**) Plotting gridlines and busbars.

**Figure 12 sensors-21-04292-f012:**
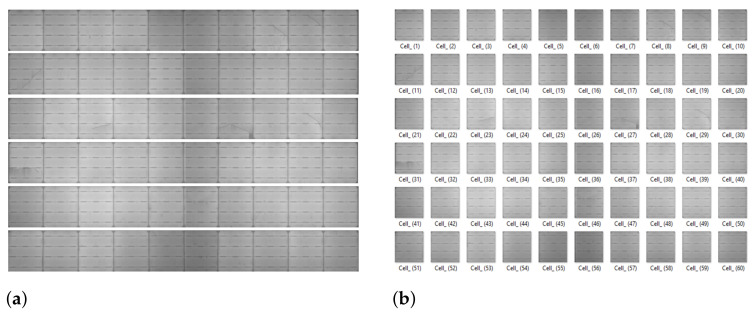
Cell segmentation. (**a**) Extracting horizontal strip regions. (**b**) Segmented cells.

**Figure 13 sensors-21-04292-f013:**
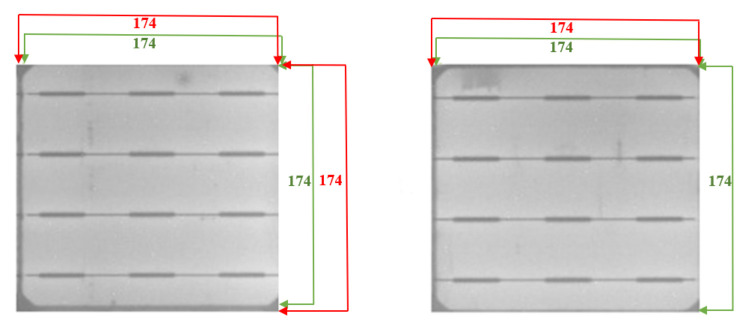
Errors in cell segmentation, where the green and red labels correspond to the true and the extracted cell boundaries, respectively.

**Figure 14 sensors-21-04292-f014:**
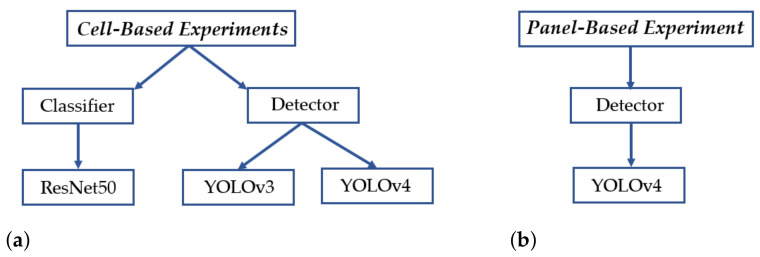
The organization of experimental comparisons. (**a**) Cell-image-based experiments. (**b**) Panelimage-based experiments.

**Figure 15 sensors-21-04292-f015:**
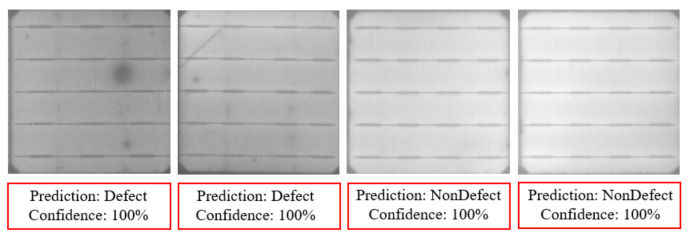
Results of cell image classification.

**Figure 16 sensors-21-04292-f016:**
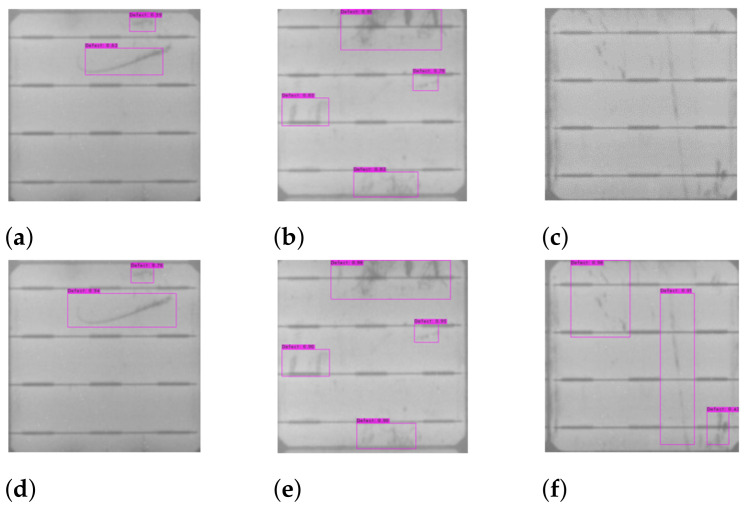
The defect detection results of cell images by (**a**–**c**) YOLOv3 and (**d**–**f**) YOLOv4.

**Figure 17 sensors-21-04292-f017:**
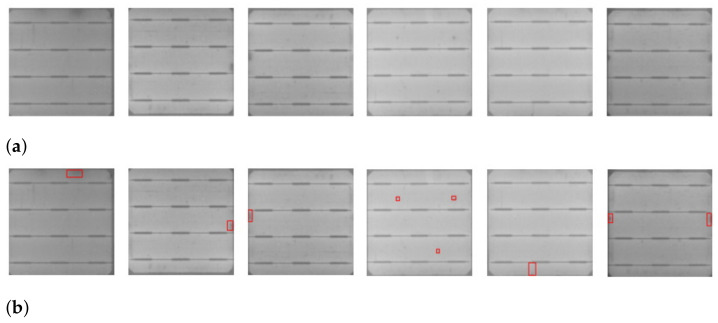
Misdetected cells with defects. (**a**) The original cell images. (**b**) Their ground truth annotations of defects.

**Figure 18 sensors-21-04292-f018:**
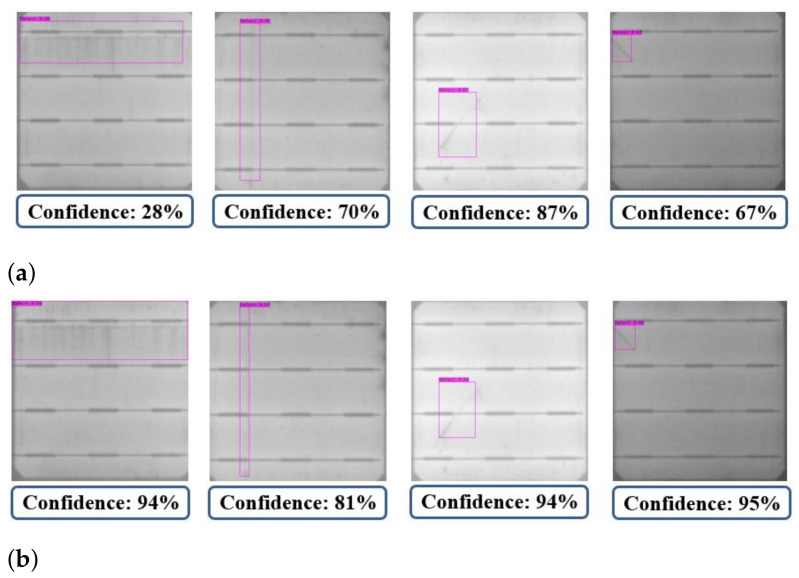
Detection results of (**a**) YOLOv3 and (**b**) YOLOv4.

**Figure 19 sensors-21-04292-f019:**
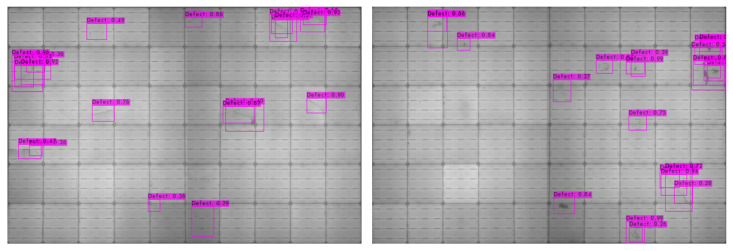
Snapshots of defect detection results of the panel-based approach.

**Figure 20 sensors-21-04292-f020:**
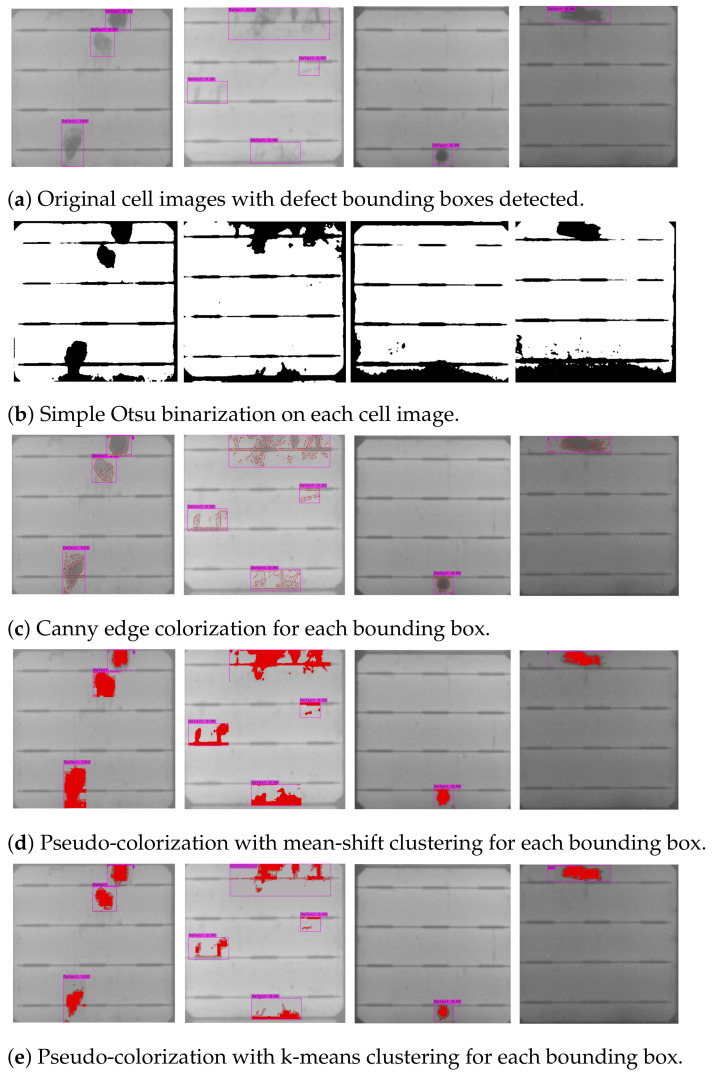
Comparisons of results of different pseudo-colorization techniques applied to (**a**) cell images with detected bounding boxes of defects. The colorization results of conventional image processing methods of the Otsu binarization on cells and of the Canny edge detection on bounding boxes are shown in (**b**,**c**), respectively. The clustering and coloring results on bounding boxes using k-means and man-shift clustering techniques are given in (**d**,**e**), respectively.

**Figure 21 sensors-21-04292-f021:**
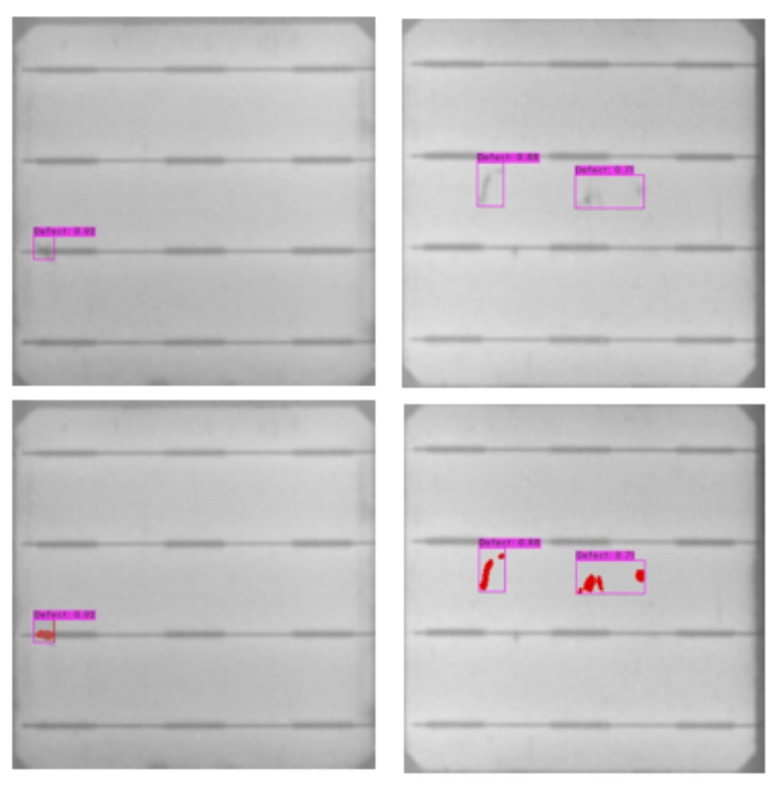
Snapshots of detected defect bounding boxes (**upper row**) that contain obscure defects and their color-marking results (**lower row**).

**Table 1 sensors-21-04292-t001:** The EL image dataset.

	Panel Images	Cell Images
Training	96	5760
Testing	23	1380

**Table 2 sensors-21-04292-t002:** The augmented cell image dataset for defect classification.

	Defect	NonDefect
Training	3537	7320
Testing	1179	2440

**Table 3 sensors-21-04292-t003:** The training and testing datasets for panel-based and cell-based defect detection.

	Panel-Based	Cell-Based
Training	96	1257
Testing	23	308

**Table 4 sensors-21-04292-t004:** The structure of ResNet50.

Layers	Image Size	
MaxPooling2D	173 × 173	
conv1	87×87	Filter size (7×7); 64 Filters; Stride 2
conv2_x	44 × 44	$\left[\begin{array}{cc} \mathrm{Filter}\mathrm{size}\;(1\times 1); & 64\;\mathrm{Filters}\\ \mathrm{Filter}\mathrm{size}\;(3\times 3); & 64\;\mathrm{Filters}\\ \mathrm{Filter}\mathrm{size}\;(1\times 1); & 256\;\mathrm{Filters} \end{array}\right]$× 3
conv3_x	22 × 22	$\left[\begin{array}{cc} \mathrm{Filter}\mathrm{size}\;(1\times 1); & 128\;\mathrm{Filters}\\ \mathrm{Filter}\mathrm{size}\;(3\times 3); & 128\;\mathrm{Filters}\\ \mathrm{Filter}\mathrm{size}\;(1\times 1); & 256\;\mathrm{Filters} \end{array}\right]$× 4
conv4_x	11 × 11	$\left[\begin{array}{cc} \mathrm{Filter}\mathrm{size}\;(1\times 1); & 256\;\mathrm{Filters}\\ \mathrm{Filter}\mathrm{size}\;(3\times 3); & 256\;\mathrm{Filters}\\ \mathrm{Filter}\mathrm{size}\;(1\times 1); & 1028\;\mathrm{Filters} \end{array}\right]$× 6
conv5_x	6 × 6	$\left[\begin{array}{cc} \mathrm{Filter}\mathrm{size}\;(1\times 1); & 512\;\mathrm{Filters}\\ \mathrm{Filter}\mathrm{size}\;(3\times 3); & 512\;\mathrm{Filters}\\ \mathrm{Filter}\mathrm{size}\;(1\times 1); & 2048\;\mathrm{Filters} \end{array}\right]$× 3
Flatten	1 × 1	
Dropout		
FC		Units: 2, Activation: softmax

**Table 5 sensors-21-04292-t005:** Confusion matrix of the test set for defect classification.

		Predicted Label	
		Defect	NonDefect
**True**	Defect	1169	10
**Label**	NonDefect	5	2435

**Table 6 sensors-21-04292-t006:** SCDD Experimental Results.

Images	Experiments		Precision (%)	Results Recall (%)	F1-Score (%)
Panel-based	Detector	YOLOv4	51.0	69.0	58.65
Cell-based	Classifier	ResNet50	99.6	99.2	99.39
	Detector	YOLOv3	79.0	76.0	77.47
		YOLOv4	**99.8**	**96.0**	**97.86**

## Data Availability

The EL images used in our research work are provided by a Solar Photovoltaic company under a private agreement.

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
