# Peer review of "Efficient Cell Segmentation from Electroluminescent Images of Single-Crystalline Silicon Photovoltaic Modules and Cell-Based Defect Identification Using Deep Learning with Pseudo-Colorization"

_sensors, 2021, doi:10.3390/s21134292_

Round 1
Reviewer 1 Report
The manuscript utilizes an automatic cells segmentation technique along with a convolutional neural network-based defect detection system with pseudo colorization of defects of single-crystalline silicon solar PV modules. A rapid EL image of the segment cells generation process takes about 2.17 s in the automatic cell segmentation methodology. The average segmentation errors along the x-direction and y-direction 18 are only 1.6 pixels and 1.4 pixels, respectively. Not only the defect detection approach on segmented cells achieves 99.8% accuracy, but pseudo colorization of defect regions enhances the visualization. Hence the manuscript can be further considered in 'Sensors'.
-Few citations are missing in this field (Review of Failures of Photovoltaic Modules - SUPSI Instory; Fault diagnosis of Photovoltaic Modules - Haque - 2019 - Energy Science & Engineering - Wiley Online Library; Evaluating the reliability of crystalline silicon photovoltaic modules in harsh environment - ScienceDirect), may need a mention.
-I-V curves of defect-free and defect-rich segments/cells are missing.
-How facile is this defect-detection system under harsh outdoor condition or the modules has to be checked in the laboratory?
Author Response
Dear Respected,
We deeply thank you and appreciate your review of our manuscript.
We have attached a response file.
Please find the attachment.
Thank you.

Reviewer 2 Report
In this paper, the authors propose an automatic cell segmentation methodology that is developed to extract cells from an electroluminescence image. In addition, the authors propose a CNN-based defect detector and can be visualized with pseudo colors. Some points should be included within the manuscript in order to improve the publication.
- For the pseudo colorization procedure, the authors use an algorithm based on k-means clustering. A comparison with other algorithms is necessary. Why the proposed method is better? It is not clear. Please compare with the other similar algorithms.
- More data must be included in Section 5. The authors should comment on their results analytically.
- In the conclusions sections, the authors should refer to the advantages and summarize the possible limitations of their proposed algorithm. They should emphasize and analyze them in detail.
- The data set which is used for training, validating, and testing the proposed CNN should be described in more detail.
- In section 5.2, the author mentioned this sentence “Even though the defect detection on a cell-based approach is time-consuming than that of a panel-based approach, we propose the implementation of cell segmentation and cell-based defect detection because defect detection is more accurate and efficient in the cell-based approach.” More justification should be furnished on this issue.
- Some assumptions are stated in various sections. More justifications should be provided on these assumptions. Evaluation on how they will affect the results should be made.
- Please introduce discussions with other articles in your conclusions. Provides three sample articles related to artificial intelligence techniques:
-
- Weipeng Wang, Shanshan TU and Xinyi Huang, “IKM-NCS: A Novel Clustering Scheme Based on Improved K-Means Algorithm”, Engineering World, Volume 1, 2019, pp. 103-108.
- Bacha Sawssen, Taouali Okba, Liouane Noureeddine, “A Mammographic Images Classification Technique via the Gaussian Radial Basis Kernel ELM and KPCA”, Int. J. of Applied Mathematics, Computational Science and Systems Engineering, Volume 2, 2020, pp. 92-98.
- Stella Vetova, “A Comparative Study of Image Classification Models using NN and Similarity Distance”, Ιnternational Journal of Electrical Engineering and Computer Science (EEACS), Volume 3, 2021, pp. 109-113.
Author Response

(The authors gave the same response as above.)

Round 2
Reviewer 2 Report
The revised version of the paper has been improved with respect to its original version and it appears to adequately address the comments and suggestions of the previous review as explained also in the Authors' responses to the reviewer comments.